# Do Rural Second Homes Shape Commensal Microbiota of Urban Dwellers? A Pilot Study among Urban Elderly in Finland

**DOI:** 10.3390/ijerph18073742

**Published:** 2021-04-02

**Authors:** Mika Saarenpää, Marja I. Roslund, Riikka Puhakka, Mira Grönroos, Anirudra Parajuli, Nan Hui, Noora Nurminen, Olli H. Laitinen, Heikki Hyöty, Ondrej Cinek, Aki Sinkkonen

**Affiliations:** 1Ecosystems and Environment Research Programme, Faculty of Biological and Environmental Sciences, University of Helsinki, Niemenkatu 73, 15140 Lahti, Finland; mika.saarenpaa@helsinki.fi (M.S.); marja.roslund@helsinki.fi (M.I.R.); riikka.puhakka@helsinki.fi (R.P.); mira.m.gronroos@helsinki.fi (M.G.); anirudra.parajuli@ki.se (A.P.); nan.hui@sjtu.edu.cn (N.H.); 2Department of Medicine, Karolinska University Hospital, Huddinge, 141 86 Stockholm, Sweden; 3School of Agriculture and Biology, Shanghai Jiao Tong University, 800 Dongchuan RD. Minhang District, Shanghai 200240, China; 4Faculty of Medicine and Health Technology, Tampere University, Arvo Ylpön katu 34, 33520 Tampere, Finland; noora.nurminen@tuni.fi (N.N.); olli.laitinen@tuni.fi (O.H.L.); heikki.hyoty@tuni.fi (H.H.); 5Second Faculty of Medicine, Charles University, V Úvalu 84, 150 06 Prague 5, Czech Republic; ondrej.cinek@lfmotol.cuni.cz; 6Natural Resources Institute Finland, Itäinen Pitkäkatu 4 A, 20520 Turku, Finland

**Keywords:** second home, cottage, outdoor recreation, gut microbiota, fecal microbiota, stool sample, elderly, immune-mediated diseases, rural areas

## Abstract

According to the hygiene and biodiversity hypotheses, increased hygiene levels and reduced contact with biodiversity can partially explain the high prevalence of immune-mediated diseases in developed countries. A disturbed commensal microbiota, especially in the gut, has been linked to multiple immune-mediated diseases. Previous studies imply that gut microbiota composition is associated with the everyday living environment and can be modified by increasing direct physical exposure to biodiverse materials. In this pilot study, the effects of rural-second-home tourism were investigated on the gut microbiota for the first time. Rural-second-home tourism, a popular form of outdoor recreation in Northern Europe, North America, and Russia, has the potential to alter the human microbiota by increasing exposure to nature and environmental microbes. The hypotheses were that the use of rural second homes is associated with differences in the gut microbiota and that the microbiota related to health benefits are more diverse or common among the rural-second-home users. Based on 16S rRNA Illumina MiSeq sequencing of stool samples from 10 urban elderly having access and 15 lacking access to a rural second home, the first hypothesis was supported: the use of rural second homes was found to be associated with lower gut microbiota diversity and RIG-I-like receptor signaling pathway levels. The second hypothesis was not supported: health-related microbiota were not more diverse or common among the second-home users. The current study encourages further research on the possible health outcomes or causes of the observed microbiological differences. Activities and diet during second-home visits, standard of equipment, surrounding environment, and length of the visits are all postulated to play a role in determining the effects of rural-second-home tourism on the gut microbiota.

## 1. Introduction

The prevalence of immune-mediated diseases, such as allergies, asthma, and autoimmune diseases, have increased significantly in developed countries over the last 30 years [1]. For example, the prevalence of asthma in developed countries can be as high as 21% [2]. The amount of urban developed area is expected to triple in size by 2030 [3], and in 2050, 70% of the world population is predicted to be living in cities [4]. With rapid urbanization, the burden of immune-mediated diseases is expected to increase abruptly in the 21st century.

Urban lifestyle is often characterized by a high level of hygiene, and according to the hygiene hypothesis and its variants (e.g., biodiversity hypothesis), decreased interaction with microbes can explain the high prevalence of immune-mediated diseases in developed countries [5,6,7,8]. Exposure to diverse microbial communities, especially in childhood, is essential for the natural development and functioning of the human immune system [6,8,9,10,11,12]. Kondrashova et al. [13] found the incidence of several immune-mediated diseases in the more urbanized Eastern Finland to be several times higher than in the adjacent Russian Karelia. Populations in both areas have a similar genetic background, and the higher incidence on the Finnish side is therefore believed to be caused by environmental factors, such as high level of hygiene and lack of daily contacts with environmental microbiota.

In addition to the lifestyle, urbanization causes changes in the environment, and urban areas tend to be biologically less diverse than the surrounding rural areas [14]. Pollution, for example, has been found to affect soil microbiota, and soil contamination has been associated with altered commensal microbiota and endocrine disruption [15,16,17]. The biodiversity hypothesis states that contact with natural environments enriches the human microbiota and thus can help prevent the development of immune-mediated diseases [7,18,19,20]. Interaction with agricultural land and forests is considered particularly important [21]. Children living on farms have been found to suffer less from asthma and atopy [10,12,21], and lower biodiversity and amount of green environment around home have been associated with atopic sensitization and altered gut microbiota composition [7,22,23]. Urbanization has decreased contact with nature and diverse environmental microbial communities [24,25]. This can lead to an imbalance in the human microbiota (known as dysbiosis) and eventually to immunological disorders [6,7,10,20].

Gut microbes are associated with multiple systemic and immunomodulatory effects and endocrine signaling pathways, and gut microbiota dysbiosis has been linked with inflammatory bowel disease (IBD), colorectal cancer, obesity, allergy, atopy, and type 1 diabetes, among others [26,27,28,29,30,31]. Different types of gut microbiota dysbiosis are characterized by distinct changes in microbiota composition. For example, decreased total bacterial diversity has been observed in IBD, psoriatic arthritis, and Sjögren’s syndrome [32,33,34], reduced complexity of the phylum Firmicutes in Crohn’s disease [35], and increased ratio of Firmicutes to Bacteroidetes in irritable bowel syndrome [36]. Changes in the butyrate-producing bacteria belonging to, for example, the families Lachnospiraceae and Ruminococcaceae are of special importance in overall gut health and in the development of numerous inflammatory conditions [37,38,39]. The elderly gut microbiota profile differs from the healthy adult one, and studies have reported a decrease in the overall diversity and species diversity for *Bacteroides*, *Prevotella*, *Bifidobacterium*, and *Lactobacillus* along with an increase in the species number within the Enterobacteriaceae, *Clostridium*, Proteobacteria, staphylococci, and streptococci [40,41,42,43].

The interaction with nature and environmental materials, for example through outdoor recreation or landscaping materials, offers a natural way of increasing exposure to diverse microbial communities. In recent studies, a diverse environmental microbiota has been incorporated into gardening and health-promoting materials to increase the microbial exposure of urban dwellers. These studies demonstrate for the first time how even short-term direct contacts with soil and plant materials affect the skin and gut microbiota and enhance immune modulation [44,45,46,47].

While deliberate direct exposure to environmental materials and microbes has been found to modify the human microbiota and immune response, there is little research on the possible effects of unimposed nature-based activities, such as outdoor recreation. Outdoor recreation refers to any leisure time activity taking place in a natural environment, such as hiking, fishing, hunting, berry and mushroom picking, camping, swimming, leisure-time gardening, bird watching, and viewing landscapes. Many of these activities increase the physical interaction with natural environments and environmental microbes, which might result in changes in the human microbiota and immune response [23,24,44,45,48].

In Finland, 96% of the adult population reports participating in outdoor activities, on average 2–3 times a week and 170 times a year [49]. Rural-second-home tourism is a popular form of outdoor recreation in Finland and other parts of Northern Europe, as well as in Russia and North America. It provides multiple psychological, social, and physiological health and well-being benefits for all age groups [50]. In Finland, over 40% of the population has been estimated to visit a rural second home regularly, and the amount of rural second homes per capita is one of the highest in the world [51,52]. As a consequence of coronavirus disease 2019 (COVID-19), second homes sales and rentals have increased in Finland and around the world, as people wish to escape the pandemic to the safety of rural second homes [53].

In Finland, visiting rural second homes has been found to increase the rate of participation in many forms of outdoor recreation, such as wild berry and mushroom picking, fishing, swimming, growing own vegetables, chopping firewood, and going for a walk [49,54]. Most Finnish rural second homes are still without an indoor flush toilet, dishwasher, shower, and other amenities, and water for bathing, dishwashing, and drinking often comes from the natural waters (e.g., well, spring, lake) [55]. Rural second home visits characterized by various outdoor activities and a lower level of hygiene have the potential to alter the human microbiota, especially among urban people who have been found to use rural second homes more often than their rural counterparts [51]. The summer vacation season is the main rural-second-home season in Finland, but the standard of equipment is increasing and allows for prolonged second-home seasons. Currently, a third of the Finnish rural second homes are suitable for winter use [55]. Visiting rural second homes during cold, snowy winters might not have the same effect on microbial exposure as in summer, as the transfer of health-related environmental bacteria indoors has been shown to be lower in winter [56].

In this study, the fecal bacterial communities (as a proxy for gut microbiota) were compared between urban elderly visiting rural second homes and elderly without regular access to a rural second home (hereafter referred to as the control group). It was tested whether the use of rural second homes is associated with changes in fecal bacterial composition. It has been shown that the everyday living environment affects commensal microbiota and that the microbiota composition differs between people living in urban and in rural environments [23,24,57]. It is not known whether visiting rural second homes is associated with gut microbiota composition. In the current study, it was hypothesized that the rural-second-home users have differences in their gut microbiota compared to the control group. It was also hypothesized that the gut microbiota associated with health benefits are more diverse or more common among rural-second-home users than in the control group.

## 2. Materials and Methods

### 2.1. Study Area and Participants

The participants—25 elderly people (65–79 years) living in urban apartment houses in the city of Lahti, Finland—were chosen from a large prospective study called GOAL (Good Aging in Lahti region, 2002–2012). The original GOAL study aimed to find connections between the living environment and chronic diseases and functional disabilities in the elderly and retired population [58]. In 2015, 60 GOAL participants were invited to participate in a follow-up study (2015–2016). In the follow-up study, stool samples for microbial analysis and survey data were collected [23,25]. Twenty-five participants from the follow-up study living in urban areas were included in this pilot study, and 10 of them were rural-second-home users. The second-home users were asked to report the number of years they had been using a second home, the number of visiting days per year, and how much time was usually spent outdoors vs. indoors (Table 1).

All participants with at least one of the following noncommunicable diseases (NCDs, also known as chronic diseases) affecting the immune response were excluded in the 2015 follow-up study: celiac disease, rheumatoid arthritis, chronic obstructive pulmonary disease, diabetes, psoriasis, multiple sclerosis, corticosteroid-treated asthma, chronic obstructive pulmonary disease, and actively treated or metastatic cancer. Daily smokers and participants with dementia or treated with corticosteroid and other immunosuppressive medication were also excluded. If the participants were treated with antibiotics within the last six months before giving the stool samples, they were excluded. None of the participants owned pets with access to the outdoors.

### 2.2. Sample Collection, DNA Extraction, Amplification, and Sequencing

In this study, stool samples collected in August 2015 were analyzed (see [23], Supplementary Figure S2 for a flowchart describing the sample collection and subsequent processes). Samples collected in August were used because the summer vacation season is the peak rural-second-home season in Finland and because exposure to environmental microbiota among aging Finns is high during the summer months [56]. Sample collection, DNA extraction, and amplicon sequencing were described in our previous study [44]. In short, participants took the samples independently using a collection kit and stored them at −20 °C until the samples were collected by the study personnel. The samples were transferred in dry ice and stored at −80 °C until further analyses. DNA was extracted from 30 to 60 mg of frozen, unprocessed stool, and bacterial community analyses were based on the amplification of the V4 region of 16S rRNA. Raw sequence reads are available in the Sequence Read Archive (https://www.ncbi.nlm.nih.gov/sra accessed on 31 March 2021) with accession numbers SAMN08991885–SAMN08992045.

### 2.3. Bioinformatics

Paired-end sequence data (.fastq) from the stool samples were processed using Mothur (version 1.39.5, http://www.mothur.org accessed on 31 March 2021) [59], as described in our earlier studies [23,44], following the protocols by Schloss and Westcott [60] and Kozich et al. [61]. The sequences were aligned using the Mothur version of SILVA bacterial reference (version 132) [62]. Less abundant (≤10 sequences across all experimental units) operational taxonomic units (OTUs) were removed to avoid PCR or sequencing artifacts [63]. Each sample was subsampled to 4024 sequences to control for the varying number of sequences.

The 16S rRNA OTU data were picked against the Greengenes Database [64] according to 97% similarity, and the OTU table was used to predict metabolic functions by referencing the Kyoto Encyclopedia of Genes and Genomes (KEGG) ortholog database (release 89.0) in PICRUSt (http://picrust.github.io/picrust/ accessed on 31 March 2021) [65].

### 2.4. Statistical Analyses

Statistical analyses were conducted for the whole bacterial community at different taxonomic levels (i.e., OTU, family, order, class, and phylum) and for the functional orthologs. At the OTU level, analyses were conducted within the most abundant dominant gut phyla and predominant colon families (Table 2); *p*-values were corrected with Benjamini–Hochberg correction to account for the multiple testing [66].

The Shannon and Simpson diversity indices, observed richness, and predicted functional orthologs of the bacterial communities were compared between the rural-second-home users and the control group using the Student’s *t*-test or Mann–Whitney U test. The Student’s *t*-test was used when the data were normally distributed based on the Shapiro–Wilk test, and the Mann–Whitney U test was used when the data were not normally distributed.

Differences in the bacterial community composition between the groups were compared using permutational multivariate analysis of variance (PERMANOVA; 999 permutations). The bacterial community compositions were visualized with non-metric multidimensional scaling (NMDS). Both PERMANOVA and NMDS were based on the Bray–Curtis dissimilarity. All statistical analyses and data visualization were performed using the R statistical software environment (version 4.0.3, R Foundation, Vienna, Austria) [68] and the packages phyloseq (version 1.32.0) [69], ggplot2 (version 3.3.2) [70], and vegan (version 2.5-6) [71].

## 3. Results

### 3.1. Characterization of the Bacterial Communities

Based on Illumina MiSeq sequencing of the bacterial 16S rRNA gene, the most common bacterial phyla in the stool samples of the rural-second-home users and the control group were Firmicutes, Bacteroidetes, Proteobacteria, and Actinobacteria (Figure 1).

### 3.2. Diversity of the Bacterial Communities

The Shannon diversity of the fecal microbiota was higher in the control group than among the rural-second-home users at the OTU, family, order, class, and phylum levels, for the phylum Firmicutes and for the families Ruminococcaceae and Lachnospiraceae (Figure 2, Table 3). For the phyla Bacteroidetes, Actinobacteria, and Proteobacteria and for the family Bacteroidaceae, no differences in Shannon diversity were observed between the rural-second-home users and the control group. The observed richness followed a similar pattern and was higher in the control group than among the rural-second-home users at the OTU, family, order, class, and phylum levels, for the phylum Firmicutes and for the family Ruminococcaceae. For the phyla Bacteroidetes, Actinobacteria, and Proteobacteria and for the families Lachnospiraceae and Bacteroidaceae, no differences in the observed richness were detected between the groups. The Simpson diversity differed from the Shannon diversity and observed richness: only the diversity of the phylum Firmicutes was observed to be higher in the control group than among the rural-second-home users, and at all other taxonomic levels, no differences were detected between the groups.

### 3.3. Bacterial Community Composition and Functional Profile

The permutational analysis of variance (PERMANOVA) revealed that the fecal bacterial community compositions did not differ between the rural-second-home users and the control group at the OTU (Figure 3) or any other taxonomic level (Table 4).

According to the predicted gut metagenomic functions, the fecal microbiota of the rural-second-home users showed lower levels of RIG-I-like receptor signaling pathway than the control group (*p* = 0.019). No other differences in the functional orthologs were observed (*p* > 0.1, except for endocytosis (*p* > 0.006); data not shown).

## 4. Discussion

In this pilot study, the fecal bacterial communities of urban elderly with and without regular access to a rural second home were compared to see whether the use of second homes is associated with changes in the gut microbiota. Previous studies have shown how the everyday living environment affects the human microbiota [7,22,24,25], but this is the first time when the effects of rural-second-home tourism, a form of outdoor recreation, on the gut microbiota have been investigated. The hypotheses were that the rural-second-home users have differences in their gut microbiota compared to the control group and that the gut microbiota associated with health benefits are more diverse or more common among the second-home users. Based on Shannon diversity and observed richness, this pilot study indicates that the fecal bacterial diversity of urban elderly rural-second-home users is lower than in the control group. Differences in the fecal bacterial diversity were observed at the OTU, family, order, class, and phylum levels, and for the OTUs, within the phylum Firmicutes and the families Lachnospiraceae and Ruminococcaceae. Interestingly, when the Simpson diversity indices were compared between the groups, differences in fecal bacterial diversity were observed only for the OTUs within the phylum Firmicutes. The Simpson diversity has been seen to emphasize the evenness of the community, whereas the Shannon diversity is more affected by the amount of rare species. The results were somewhat surprising, as previous studies have shown that direct contact with nature diversifies the gut microbiota [44,47]. The researchers originally expected that rural-second-home users regularly engage in activities that increase exposure to nature. A diverse gut microbiota is often associated with health benefits and eubiosis. While the observed differences in α-diversity confirm that the gut microbiota of the rural-second-home users is different compared to that of the control group, the microbiota associated with health benefits (Lachnospiraceae and Ruminococcaceae) were not more diverse or common among the second-home users. Furthermore, the bacterial community composition did not differ between the rural-second-home users and the control group.

Previous studies have found that the gut microbiota of elderly people with health issues is characterized by, for example, a reduction in the overall diversity, Bacteroides-Prevotella group, bifidobacteria, lactobacilli, and some clostridia, and an increase in enterobacteria, proteobacteria, staphylococci, and streptococci [72,73,74]. A reduction in the phylum Firmicutes, a major gut phylum, has also been observed in the elderly [75]. The reduction of Firmicutes has also been linked to colorectal cancer [76], type 2 diabetes [77], and Crohn’s disease [35]. On the other hand, elevated Firmicutes levels, especially of bacteria belonging to the family Lachnospiraceae, have been associated with irritable bowel syndrome (IBS) [78]. The families Ruminococcaceae and Lachnospiraceae, both belonging to the phylum Firmicutes, contain many species that produce butyrate, an important bacterial metabolite with positive anti-inflammatory, epithelial barrier-protective, and cell-regulatory effects [38]. The decreased abundance of both Ruminococcaceae and Lachnospiraceae has been linked with microbial dysbiosis in IBD [27,76,79] and even with a higher risk of stroke [80]. A reduction in the overall diversity of the gut microbiota has also been observed in IBD [32], as well as in psoriatic arthritis and Sjögren syndrome [33,34].

Based on the predicted gut metagenomic functions, the RIG-I-like receptor signaling pathway was lower among the rural-second-home users than in the control group. RIG-I-like receptors are part of the eukaryotic intracellular pattern recognition, and bacterial and viral infections are known to activate the RIG signaling pathway [81,82]. The results do not suggest that the detected bacteria express RIG-I-like receptor signaling pathway genes but that they might express molecules that could act as part of the pathway. The effects of RIG-I-like receptors are mediated by the induction of interferons [83]. As RIG-I-like receptors are able to induce a strong proinflammatory response through interferon-stimulated elements, this pathway is tightly regulated [84,85] to avoid inflammation-associated overreactions. Intriguingly, RIG-I-mediated interferon responses have been detected without viral or bacterial triggers in association with autoimmune diseases [86], such as relapsing–remitting multiple sclerosis, Rheumatoid arthritis, Sjögren’s syndrome, and systemic lupus erythematosus [87,88,89,90,91]. The current results suggest that the RIG-I-like receptors in elderly spending time at rural second homes might be better regulated, leading to a lower risk of immune-mediated diseases, such as allergies [22].

Since specific changes in the gut microbiota can have both health- and illness-related consequences, it is difficult to determine whether the observed differences are associated with health outcomes. This is even more true in the case of the elderly, whose gut microbiota display greater inter-individual variability than that of younger adults [92]. This variability is caused by many factors, such as changes in digestion, bowel function, medication, everyday living environment, mobility, and, perhaps most importantly, diet [41,42,43,92]. Patterns in the microbiota composition also differ between geographical regions, which highlights the danger of generalizing health-related observations in the elderly gut microbiota [42,43,93]. Because defining a healthy elderly gut microbiota is challenging, it cannot be concluded that the lower gut microbiota diversity of rural-second-home users is a sign of gut dysbiosis, especially since the community composition did not differ between the studied groups. Rural-second-home tourism comes in many forms, and it can be postulated that there is a myriad of factors that determine the effects of rural-second-home tourism on the gut microbiota.

In a previous study with the same study population and stool samples, the effects of ethnicity, socioeconomic status, high parasite load, access to modern health care, and use of antibiotics were ruled out, and the importance of the everyday interaction with natural environments in the homeostasis of elderly gut microbiota was underlined (see [23], Supplementary Table S1). The type and quality of the environment surrounding homes are important determinants of the commensal microbiota [22,24,48], and for example, broad-leaved forests and diverse yard vegetation have health-related effects on the gut microbiota [17,23]. Unfortunately, information on the surrounding environment of the second homes was not collected in this study. Therefore, the details of the non-microbial diversity surrounding the second homes are unknown. However, the coverage of built area surrounding the rural second homes can be assumed to be low, i.e., resembling the land cover structure in sparsely populated rural areas [23,25]. Information on the activities during the second home visits was also missing. However, most second-home users did report that they tend to spend more time outdoors while visiting the second homes (Table 1). Having direct physical contact with nature (e.g., chopping firewood, picking wild berries or mushrooms, swimming, or gardening) probably has a different effect on the commensal microbiota than sitting on a patio or going for a walk.

While no information on the standard of equipment of the second homes is available, the equipment plausibly varies based on the reported number of visiting days. On average, rural-second-home users reported spending 67 days a year at their second homes (Table 1), but the timing and exact number of visiting days in 2015 is unknown. Those rural second homes where study participants reported spending 100–150 days per year are probably fit for winter use and resemble ordinary detached homes. However, most rural second homes in Finland, and presumably also in the current study, are typical lakeside summer cottages with latrines and no running water [51,55]. The standard of equipment can be assumed to affect the exposure to microbes through differing activities and levels of hygiene. For example, if the second home lacks electricity and electrical appliances, many activities and chores might take place outdoors (e.g., open-fire grilling). The lack of running tap water often means that water for bathing, washing dishes, and drinking comes from natural waters. Natural water sources often contain more organic material than chlorinated tap water, which can affect human exposure to microbes [94]. As washing dishes by hand is associated with a low incidence of allergic diseases [95], contact with natural water sources can be expected to have a similar health-related effect.

The effect of diet on the elderly gut microbiota composition was ruled out in a previous study with the same study population [23]. Therefore, information on the dietary habits during the second-home visits was not collected in this study. In the previous study [23], diet-related questions targeted only the everyday dietary habits, and these habits might be different during rural-second-home visits due to possibly more rudimentary conditions. While some studies have observed that rural-second-home users favor homegrown and local products [96], heavily grilled food (e.g., sausages) and alcohol are also an important component of the rural-second-home life for many [50]. The elderly gut microbiota has been strongly linked with diet [92], and diets rich in fat and poor in dietary fiber along with overconsumption of alcohol have been associated with disadvantageous changes in the gut microbiota [97,98,99,100,101,102]. Interestingly, butyrate-producing Firmicutes, such as Lachnospiraceae, are sensitive to changes in dietary fiber [103]. Dietary choices at rural second homes might be imbalanced and non-diversified due to, for example, long distances to grocery stores.

While lower gut microbiota diversity and RIG-I-like receptor signaling pathway levels were observed in urban elderly rural-second-home users, the small sample size limits the possibilities to draw conclusions on the nature or causes of these differences. Arguably, rural-second-home tourism is not automatically associated with only health- or illness-related changes in the gut microbiota. Instead, individual choices concerning diet and activities are potentially crucial explanatory factors. There are as many ways of using rural second homes as there are users. Therefore, larger studies with different age groups and geographical regions are needed. It would also be advisable to compare the fecal bacterial communities before and after the rural-second-home visits. To overcome the uncertainties of the current findings, future research should take into account the standard of equipment of the rural second homes, land cover and biodiversity around the homes, and dietary habits during the second-home visits.

## 5. Conclusions

It was hypothesized that there are differences in the gut microbiota between rural-second-home users and control group and that microbiota associated with health benefits are more diverse or common among rural-second-home users. The findings of this pilot study suggest that there are differences in the gut microbiota: the use of rural second homes is associated with lower gut microbiota diversity and RIG-I-like receptor signaling pathway levels when compared with non-use. However, the microbiota associated with health benefits were not more diverse or common among the rural-second-home users. Although the observed differences have been associated with health outcomes in previous studies, making any conclusions related to these is not meaningful in the context of this study. Larger studies taking into account the standard of equipment of the rural second homes, the type and quality of the surrounding environment, the choice of activities, the length and season of the visits, and dietary habits are encouraged to understand the role of rural-second-home tourism in shaping the gut microbiota structure and function and their potential association with health.

## Figures and Tables

**Figure 1 ijerph-18-03742-f001:**
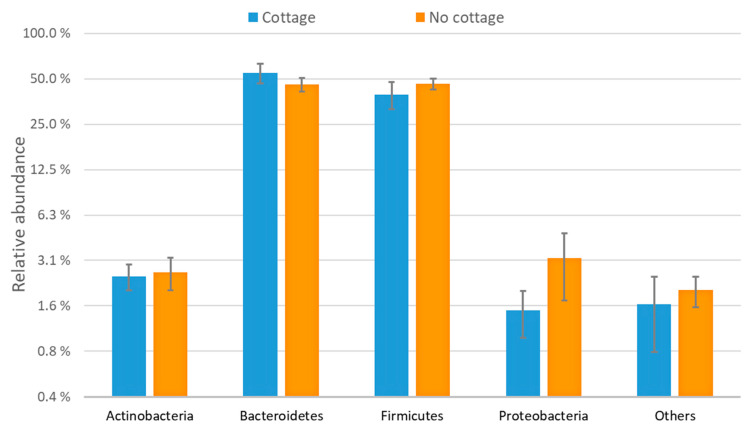
Relative abundances of bacterial phyla in stool samples of 10 rural-second-home users (cottage; blue color) and 15 control group participants (no cottage; orange color) (values are expressed as mean ± SE).

**Figure 2 ijerph-18-03742-f002:**
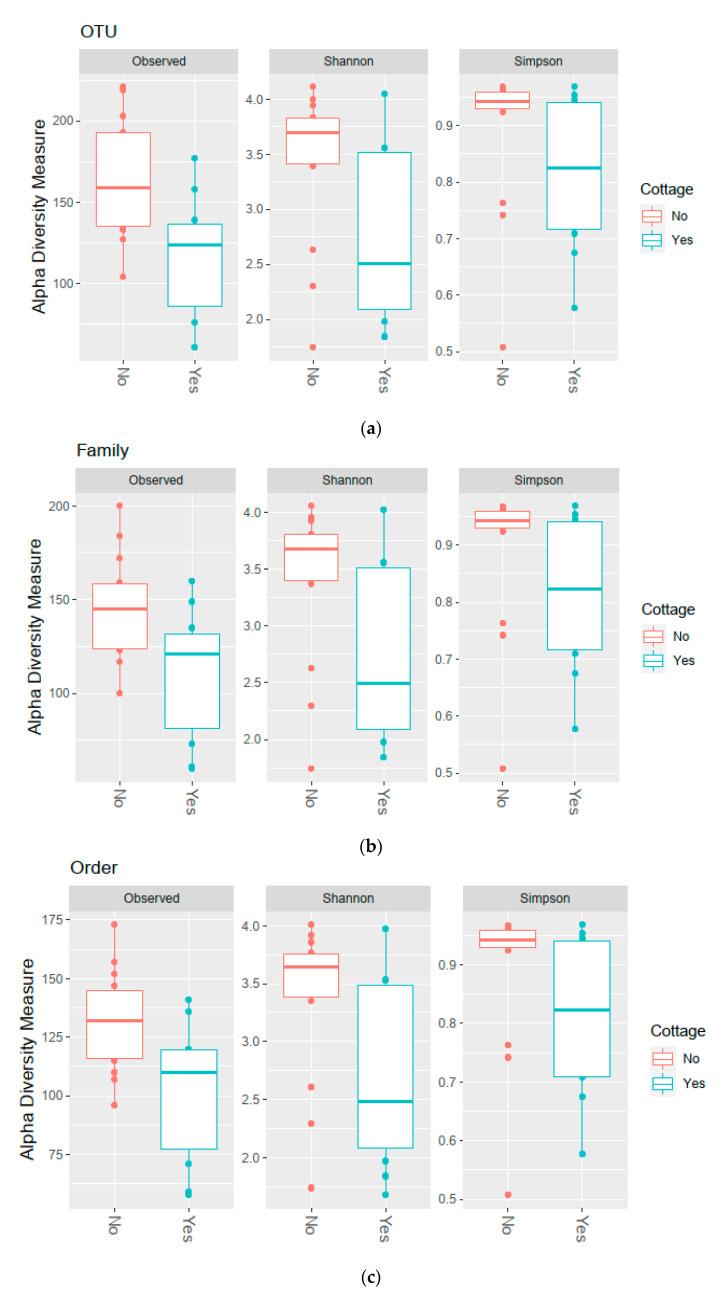
The Shannon diversity of fecal microbiota was higher in the control group (no cottage; red color) than among rural-second-home users (cottage; blue color) at all tested taxonomic levels (operational taxonomic units (OTU) (**a**), Family (**b**), Order (**c**), Class (**d**), Phylum (**e**) and for OTUs within the phylum Firmicutes (**f**) and the families Lachnospiraceae (**g**) and Ruminococcaceae (**h**). The observed richness was higher in the control group than among rural-second-home users at all tested taxonomic levels and within the phylum Firmicutes and the family Ruminococcaceae. Simpson diversity was higher in the control group only within the phylum Firmicutes. Boxplots show medians (thick line), upper and lower hinges (box), values 1.5 times IQR (whiskers), and values outside hinges (data points, outliers).

**Figure 3 ijerph-18-03742-f003:**
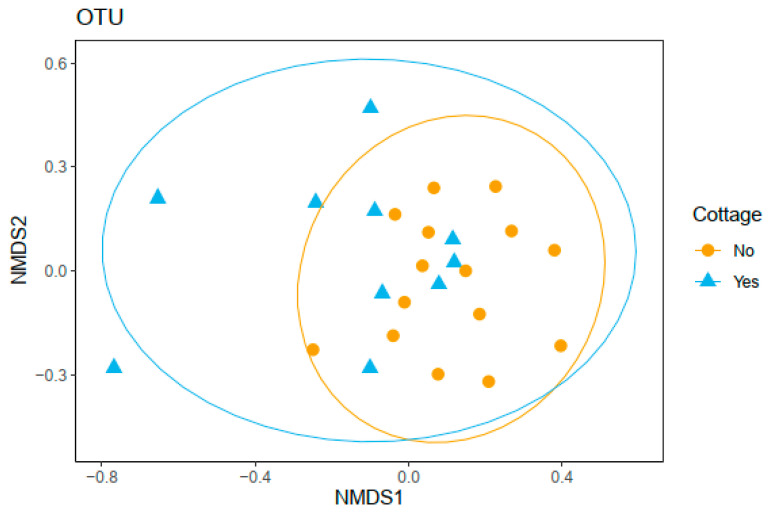
Fecal bacterial community composition did not differ between rural-second-home users and control group (*p* = 0.139, stress = 0.178, *F* = 1.327, *R*^2^ = 0.055). Non-metric dimensional scaling ordination based on Bray–Curtis dissimilarity of bacteria in stool samples of urban rural-second-home users (cottage; blue triangles) and control group (no cottage; orange dots) at the operational taxonomic unit (OTU) level is shown.

**Table 1 ijerph-18-03742-t001:** Rural second homes had been used on average for 36 years, and the number of visiting days per year was on average 67. Most participants reported spending more time outdoors than indoors while visiting rural second homes.

Years Used	Visiting Days per Year	Time Spent Outdoors vs. Indoors
46	40	Outdoors
23	10	Outdoors
30	35	Outdoors
27	90	Equally
-	100	Outdoors
44	30	Outdoors
29	150	Equally
53	30	Outdoors
61	-	Outdoors
9	120	Equally

**Table 2 ijerph-18-03742-t002:** Dominant bacterial phyla in the gut and predominant bacterial families in the colon [67].

Taxa	Role and Geography	Note
Bacteroidetes	Dominant phylum in the gut	
Firmicutes	Dominant phylum in the gut	
Actinobacteria	Dominant phylum in the gut	
Proteobacteria	Dominant phylum in the gut	
Verrucomicrobia	Dominant phylum in the gut	Not analyzed due to frequent zero-valued observations
Ruminococcaceae	Predominant family in the inter-fold regions of the lumen	
Lachnospiraceae	Predominant family in the inter-fold regions of the lumen	
Bacteroidaceae	Predominant family in the digesta	
Prevotellaceae	Predominant family in the digesta	Not analyzed due to frequent zero-valued observations
Rikenellaceae	Predominant family in the digesta	Not analyzed due to frequent zero-valued observations

**Table 3 ijerph-18-03742-t003:** Summary of bacterial diversity results. *p*-values, Benjamini–Hochberg critical values, significance after the correction, and type of the statistical test used (U or *t*-test) are given for each group.

Diversity Measure	Level	*p*-Value	Benjamini–Hochberg Critical Value [66]	Significant with FDR 0.1	Test
Shannon	Phyl. Firmicutes	0.001	0.008	Yes	U test
	Fam. Ruminococcaceae	0.014	0.017	Yes	*t*-test
	Fam. Lachnospiraceae	0.017	0.025	Yes	U test
	Phylum	0.031	0.033	Yes	U test
	Class	0.031	0.033	Yes	U test
	Order	0.031	0.033	Yes	U test
	OTU	0.035	0.042	Yes	U test
	Family	0.035	0.042	Yes	U test
	Phyl. Proteobacteria	0.154	0.050	No	*t*-test
	Phyl. Bacteroidetes	0.229	0.058	No	*t*-test
	Fam. Bacteroidaceae	0.542	0.067	No	U test
	Phyl. Actinobacteria	0.868	0.075	No	*t*-test
Simpson	Phyl. Firmicutes	0.004	0.008	Yes	U test
	Fam. Ruminococcaceae	0.027	0.017	No	U test
	Fam. Lachnospiraceae	0.031	0.025	No	U test
	OTU	0.120	0.033	No	U test
	Phylum	0.120	0.033	No	U test
	Class	0.120	0.033	No	U test
	Order	0.120	0.033	No	U test
	Family	0.120	0.033	No	U test
	Phyl. Bacteroidetes	0.267	0.042	No	U test
	Fam. Bacteroidaceae	0.579	0.050	No	U test
	Phyl. Proteobacteria	0.680	0.058	No	*t*-test
	Phyl. Actinobacteria	0.868	0.067	No	U test
Observed richness	Fam. Ruminococcaceae	0.003	0.008	Yes	*t*-test
	OTU	0.008	0.017	Yes	*t*-test
	Phyl. Firmicutes	0.009	0.025	Yes	*t*-test
	Family	0.018	0.033	Yes	*t*-test
	Class	0.019	0.042	Yes	*t*-test
	Order	0.019	0.05	Yes	*t*-test
	Phylum	0.024	0.058	Yes	*t*-test
	Phyl. Proteobacteria	0.053	0.067	No	*t*-test
	Fam. Lachnospiraceae	0.126	0.075	No	U test
	Phyl. Actinobacteria	0.548	0.083	No	*t*-test
	Phyl. Bacteroidetes	0.760	0.092	No	U test
	Fam. Bacteroidaceae	0.889	0.1	No	*t*-test

**Table 4 ijerph-18-03742-t004:** Summary of results from PERMANOVA analysis of fecal bacterial community composition.

Level	*p*-Value	*F*	*R* ^2^
Phyl. Proteobacteria	0.068	1.441	0.059
Phyl. Actinobacteria	0.080	1.661	0.067
OTU	0.139	1.327	0.055
Fam. Bacteroidaceae	0.172	1.376	0.056
Phyl. Bacteroidetes	0.254	1.205	0.050
Phylum	0.300	1.117	0.046
Order	0.318	1.088	0.045
Class	0.324	1.089	0.045
Fam. Lachnospiraceae	0.328	1.104	0.046
Family	0.343	1.071	0.044
Fam. Ruminococcaceae	0.408	0.990	0.041
Phyl. Firmicutes	0.427	1.024	0.043

## Data Availability

Raw sequence reads are available in the Sequence Read Archive (https://www.ncbi.nlm.nih.gov/sra, accessed on 31 March 2021) with accession numbers SAMN08991885-SAMN08992045.

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
