# Peer review of "Do Rural Second Homes Shape Commensal Microbiota of Urban Dwellers? A Pilot Study among Urban Elderly in Finland"

_ijerph, 2021, doi:10.3390/ijerph18073742_

Round 1
Reviewer 1 Report
This is a very well-written research paper. I appreciate the authors' extensive review of the body of knowledge and the work required to study this important topic. The acknowledgements and statements to ensure respect for individuals, fairness, and justice, along with the overall commitment to ethical research, data security and confidentiality are truly notable.
Comments
1) The paper identifies two hypotheses. The data is analyzed but I did not find the level of significance to reject each null hypothesis. Also, please address each hypothesis and the outcomes in the results/discussion and conclusion. While the findings are summarized in general, the paper would be strengthened by having subsections for each hypothesis.
2) The abstract and conclusion communicate a finding that is an outcome but it does not state the outcomes from hypothesis testing. The paper should state that the data did not support the hypotheses of the paper, however, one finding suggesting _______ and should be further investigated. As currently is written, it implies that the outcome supports one or both of the hypotheses.
3) While adding the sentence in the discussion that starts with" Interestingly ... children," certainly is valid information, it reads like a casual comment or as an opinion. The paper should communicate objective facts and information, especially in the discussion. Consider moving this information to the background.
Minor Comments
1) Eliminate "we" and/or "our" - lines 24 (abstract), 89, 127, 134, 135, 140 165, 172, 191, 197, 259, 273, 318, 324, 330, 332, 358, 371
2) Eliminate split infinitives - 161 (correct to also were), 303 (correct to strictly is regulated), 336 (correct to also are), 363 ( correct to also are)
3) Table 2 - align the first note to the correct place on the table
4) Line 171 - sentence begins with the word - "Shortly" - this needs to be rewritten as it is unclear what "shortly" means in this context, I am not sure this is the correct word to use
5) Line 177 - needs a ")" at the end of the sentence
6) Line 273 - the sentence does not make sense and needs to be rewritten to improve the clarity of the idea being conveyed
7) Line 275 - this is an awkward sentence, and needs to be rewritten
Reviewer 2 Report
The authors investigated the effects of rural second home tourism on the gut microbiome. They hypothesized that the use of rural second homes is associated with increased diversity of microbiota and changes of bacterial compositions. Based on the 16Sr RNA sequening, they analyzed the fecal microbiome from 10 rural second home users and 15 non-users. Unexpectedly, they found that the use of rural second homes is associated with lower gut microbiota diversity and there are no changes of gut microbial composition. Although their results did not support the hypothesis, the study is relatively well conducted and it encourages further research. Below are the reviewer’s comments.
- The authors listed both SILVA and Greengenes Database as bacterial references. Did they use both in the same pipeline? Please explain the point.
- The PERNOVA test was used to statistically compare the gut microbiota composition. What statistical method did you used to analyze predicted functions by PICRUSt?
- As the authors discussed, elderly people show greater inter-individual variability of gut microbiome. The reviewer would suggest to compare gut microbiota before and after rural second home tourism in the future study.
Reviewer 3 Report
This is an interesting study on the effect of rural second home recreation and activities on the gut microbiota of urban elderly in Finland. The fecal bacterial composition of stool samples were examined to see if visiting rural second homes has an influence on gut microbial biodiversity.
The study background is relevant, the methods are thoroughly described, the results are clearly displayed and the conclusions link to the obtained results.
However, the study design could have been improved. No information is provided on the eating habits of the study participants, especially during the sampling period. Diet can have a significant effect on stool and also gut microbiota.
Furthermore, no information is provided on the specific time of the year when the participants visited their rural second homes. As stated in the paper, the temperature (season) can have a significant effect on the spreading of bacteria from the rural environment to the participant.
I see that you mentioned these shortcomings in the conclusion section. Therefore, I suggest adding these parameters to make the study more scientifically sound.
Round 2
Reviewer 3 Report
Thank you for addressing my concerns and requests so accurately.
I approve your paper for publication.